# Communication-efficient algorithms for solving pressure Poisson equation for multiphase flows using parallel computers

**Soumyadip Ghosh**[1], **Jiacai Lu**[2], **Vijay Gupta**[3]*, **Gretar Tryggvason**[2]

**1** Intel Corporation, University of Notre Dame, Notre Dame, IN, United States of America, **2** Department of Mechanical Engineering, Johns Hopkins University, Baltimore, MD, United States of America, **3** School of Electrical and Computer Engineering, Purdue University, West Lafayette, IN, United States of America

* gupta869@purdue.edu

**Data Availability Statement:** Our codes are available at https://github.com/soumyadipghosh/eventpde.

## Abstract

Numerical solution of partial differential equations on parallel computers using domain decomposition usually requires synchronization and communication among the processors. These operations often have a significant overhead in terms of time and energy. In this paper, we propose communication-efficient parallel algorithms for solving partial differential equations that alleviate this overhead. First, we describe an asynchronous algorithm that removes the requirement of synchronization and checks for termination in a distributed fashion while maintaining the provision to restart iterations if necessary. Then, we build on the asynchronous algorithm to propose an *event-triggered* communication algorithm that communicates the boundary values to neighboring processors only at certain iterations, thereby reducing the number of messages while maintaining similar accuracy of solution. We demonstrate our algorithms on a successive over-relaxation solver for the pressure Poisson equation arising from variable density incompressible multiphase flows in 3-D and show that our algorithms improve time and energy efficiency.

## 1 Introduction

In this paper, we propose efficient communication strategies for solving partial differential equations (PDEs) using parallel computers. For concreteness, we focus on the pressure Poisson PDE that arises from multiphase flows that are found in a wide range of applications, including bubble columns in the chemical industry, nuclear reactors, and various aspects of metal processing. Various strategies to model such flows have been discussed [1–4]. Solving the pressure Poisson equation is usually the most time-consuming part of the numerical solution of the equations governing incompressible flows. The equations are usually discretized to form a linear system of equations. While for unsteady single phase flow it is, at least in principle, possible to invert the coefficient matrix once and then use it at every time step, in multiphase flows with time evolving phase boundaries, the density distribution and the coefficients change at every time step, thus requiring the full pressure equation to be solved repeatedly. In most cases, the linear system of equations is, thus, solved by using an iterative method. It is

**Funding:** This research was supported in part by the University of Notre Dame Center for Research Computing through its computing resources. The work of the authors was supported in part by National Science Foundation though grants CBET-1953090 and CBET-1953082. The funders had no role in study design, data collection and analysis, decision to publish, or preparation of the manuscript.

**Competing interests:** The authors have declared that no competing interests exist.

important to note that it is only the converged solution that is of interest and that convergence is usually evaluated by monitoring the residual. The solution during the intermediate iterations is of no direct relevance and can take any value consistent with driving the solution to the converged value. Ideally, this should be done as *efficiently* (in the sense of time and energy consumption) as possible.

The pressure Poisson equation falls in the broad class of elliptic PDEs. Development of strategies to improve the convergence rate of iterative methods for such PDEs has a long and illustrious history, that includes Gauss-Seidel and successive over-relaxation (SOR) methods to improve the Jacobi method and then further sophistication with alternating direction implicit (ADI), Krylov, and multigrid methods. In some cases, it is possible to use the structure of the particular problem under consideration to improve the solution strategy, such as through extrapolation [5, 6] for pressure equations for multiphase flows in which the density of one fluid is much less than the other. For the solution of these PDEs on parallel computers consisting of many processing elements (PEs), the ability to decompose the domain and solve different parts of the domain on separate PEs is essential to scaling up the calculations to problems of modern interest. Several authors have discussed parallel strategies for solving elliptic problems [7–9]. When implemented on parallel computers, all these methods generally assume full communications at every iteration and synchronized processing by all the PEs. This typically leads to significant time and communication overhead. It has been observed that the communication between elements is generally slow compared to computation done on each PE and also consumes significant energy [10]. Further communication can lead to congestion in the high performance computing (HPC) interconnects [11]. Finding ways to reduce communication is, thus, becoming increasingly important.

To tackle this issue, many approaches have been proposed in the parallel computing literature. A major direction is that of developing *asynchronous* algorithms in which communication happens as before, but without the concomitant synchronization, so the PEs do not wait for values from each other but continue their computations with whatever values were last received [12–16]. Another approach is to completely avoid communication at certain iterations, thus reducing the requirement of synchronization as well. In addition to reducing synchronization overhead as in asynchronous algorithms, such *communication-avoiding* algorithms reduce the number of messages as well. Several works have focused on relaxing the global communication needed for calculation of the basis vectors in Krylov subspace methods. As a representative example, in *s*-step methods, such communication is done once every *s* steps [17–19]. However, these *s*-step methods considered parallelization using operator decomposition, i.e., parallelization of operators like matrix-vector or matrix-matrix multiplications. This is different from the parallelization using domain decomposition considered in this paper where the entire simulation grid is divided among multiple PEs instead of the operators involved.

In preliminary work [20], we showed that triggering communication based on *events* using a simple threshold can lead to some communication savings for a simple Poisson problem resulting from electrostatics. Here, we first develop an asynchronous communication algorithm for the more complicated, but well-known pressure Poisson PDE from fluid dynamics and show that it significantly reduces the computation time. Then, we extend our previous event-triggered algorithm [20] to include a more sophisticated mechanism of triggering events based on adaptive thresholds for the fluids PDE. This leads to further savings in time and a prominent reduction in the number of messages exchanged between PEs. Such event-triggered communication has also been shown to be useful in the different context of parallel machine learning [21].

The main contribution in this paper is the design of communication strategies to accelerate iterative solutions of the non-separable pressure equation found in simulations of unsteady incompressible multiphase flows by reducing synchronization and communication. We first use asynchronous communications implemented using one-sided communication routines of the message passing interface (MPI). Not only is the local communication of boundary values with neighbors done asynchronously, but also the convergence detection is done in a distributed manner using asynchronous routines. Modern solver for elliptic equations, like the pressure Poisson equation, have reached a high degree of sophistication and their implementation on parallel computers is fairly elaborate. However, to focus on the communication aspects and to keep the solver as simple as possible we have elected to work with a very simple SOR solver. While state-of-the-art Krylov or multigrid solvers have mostly replaced SOR in computational fluid dynamics and other applications, it is introduced in almost all modern textbooks and its simple structure makes it ideal for proof of concept software implementations. As such, its performance and properties continue to be of some interest [22, 23]. We note that although several variants of the original SOR algorithm have been introduced, here we use the very basic version. This asynchronous solver is shown to be around 6 times faster than the synchronous solver for our example problem. Further, we modify the asynchronous algorithm to describe another algorithm where the communication of boundary values with neighbors happens only when certain criteria have been met, i.e., in an *event-triggered* fashion. This algorithm can reduce the number of messages communicated among the PEs by upto 90% while preserving the same level of accuracy of solution. Since number of messages is a measure of the overall volume of communication, decreasing that will alleviate the overhead associated with communication. Our codes are available at https://github.com/soumyadipghosh/eventpde.

The paper is organized as follows. Section 2 introduces the pressure Poisson equation for multiphase flows which is the PDE we use throughout the paper. Section 3 reviews the usual synchronous solver. Section 4 describes the asynchronous solver. In section 5, we extend the solver by adding event-triggered communication. In Section 6, we present results for the respective algorithms. Finally, we conclude with a discussion in Section 7.

## 2 The pressure Poisson equation for multiphase flows

The most common approach for simulations of multiphase flows is the use of the "one-fluid" formulation of the Navier-Stokes equations, where one set of equations is solved for the whole flow field on a fixed structured grid, and the motion of the different phases is tracked by advecting a marker or index function. The different phases have different material properties, including densities, and this makes the pressure equation that must be solved for incompressible flows significantly different than for single phase flow, due to the discontinuous density field. When a projection method is used to advance the solution, we first update the solution ignoring the pressure (or using the pressure field from the last time step as an approximation) and then find the pressure needed to make the new velocity field incompressible, thus projecting the velocity field on a subspace representing divergence free flows. The pressure equation can be written as

$$\nabla \cdot \frac{1}{\rho} \nabla p = \frac{1}{\Delta t} \nabla \cdot \mathbf{u}^*, \tag{1}$$

where the right hand side is the divergence of the velocity after the prediction step, $\rho$ is the discontinuous density field, and $\Delta t$ is the size of the time step used to update the momentum equations (assumed to be given). The discrete version for a regular structured staggered grid

can be expressed as follows:

$$\frac{1}{\Delta x^2} \left( \frac{p_{i+1,j} - p_{i,j}}{\rho_{i+1,j}^{n+1} + \rho_{i,j}^{n+1}} - \frac{p_{i,j} - p_{i-1,j}}{\rho_{i,j}^{n+1} + \rho_{i-1,j}^{n+1}} \right) + \tag{2}$$

$$\frac{1}{\Delta y^2} \left( \frac{p_{i,j+1} - p_{i,j}}{\rho_{i,j+1}^{n+1} + \rho_{i,j}^{n+1}} - \frac{p_{i,j} - p_{i,j-1}}{\rho_{i,j}^{n+1} + \rho_{i,j-1}^{n+1}} \right) = S_{i,j}, \tag{3}$$

where

$$S_{i,j} = \frac{1}{2\Delta t} \left( \frac{u_{i+1/2,j}^* - u_{i-1/2,j}^*}{\Delta x} + \frac{v_{i,j+1/2}^* - v_{i,j-1/2}^*}{\Delta y} \right), \tag{4}$$

assuming two-dimensional flow for simplicity and using half "integers" to indicate where the variables are on the staggered grid [24]. $\Delta x$ and $\Delta y$ are the grid line spacing in the $x$ and the $y$-direction. Since the interface separating the different fluids usually moves, the coefficients change. In addition to the discontinuous coefficients, the pressure itself is often discontinuous, if surface tension is non-zero.

The pressure equation can be solved in a number of ways such as by direct or iterative solvers. Iterative solvers are more common and many sophisticated solvers such as multigrid [25] have been implemented in widely available software packages. The Hypre library [26], for example, implements a multigrid solver that is often used to solve (2). In this paper, where we are focusing on the communications between PEs, we consider a simple parallel iterative SOR solver using domain decomposition to demonstrate the algorithms. Thus, we rewrite Eq (3) as:

$$
\begin{aligned}
p_{i,j}^{\alpha+1} = \beta & \left[ \frac{1}{\Delta x^2} \left( \frac{1}{\rho_{i+1,j}^{n+1} + \rho_{i,j}^{n+1}} + \frac{1}{\rho_{i,j}^{n+1} + \rho_{i-1,j}^{n+1}} \right) \right. \\
& \left. + \frac{1}{\Delta y^2} \left( \frac{1}{\rho_{i,j+1}^{n+1} + \rho_{i,j}^{n+1}} + \frac{1}{\rho_{i,j}^{n+1} + \rho_{i,j-1}^{n+1}} \right) \right]^{-1} \\
& \left[ \frac{1}{\Delta x^2} \left( \frac{p_{i+1,j}^{\alpha}}{\rho_{i+1,j}^{n+1} + \rho_{i,j}^{n+1}} + \frac{p_{i-1,j}^{\alpha+1}}{\rho_{i,j}^{n+1} + \rho_{i-1,j}^{n+1}} \right) + \right. \\
& \left. \frac{1}{\Delta y^2} \left( \frac{p_{i,j+1}^{\alpha}}{\rho_{i,j+1}^{n+1} + \rho_{i,j}^{n+1}} + \frac{p_{i,j-1}^{\alpha+1}}{\rho_{i,j}^{n+1} + \rho_{i,j-1}^{n+1}} \right) - S_{i,j} \right] + (1-\beta)p_{i,j}^{\alpha}.
\end{aligned}
\tag{5}
$$

Here, the subscript $\alpha$ is the iteration number and $\beta$ is the over-relaxation parameter. Although Eqs (3) and (5) are written for a 2-D flow, we solve the pressure Poisson equation for multi-phase flows in the 3-D domain shown in Fig 1. The source term $S_{i,j}$ is computed by taking one step using a full flow solver, with $\Delta t = 4 \times 10^{-6}$. Table 1 provides further details about the domain parameters. Here, we take the density of the heavy fluid to be 10, 000 times larger than the density of the lighter fluid to make the solution more challenging, since such large difference generally require a considerably larger number of iterations, compared with density ratios of $O(10 - 100)$. In the simulations reported here, we use $\beta = 1.2$. The domain decomposition is done by slicing the domain in the long dimension and using one ghost layer for each domain boundary. For the simulations, we use an HPC cluster of nodes with each node having 2 CPU Sockets of AMD's EPYC 24-core 2.3 GHz processor and 128 GB RAM per node. In order to ensure the 200 PEs for our simulations are equally populated among the 48 core AMD nodes for load balancing purposes, we use only 40 cores per node and a total of 5 such nodes. The

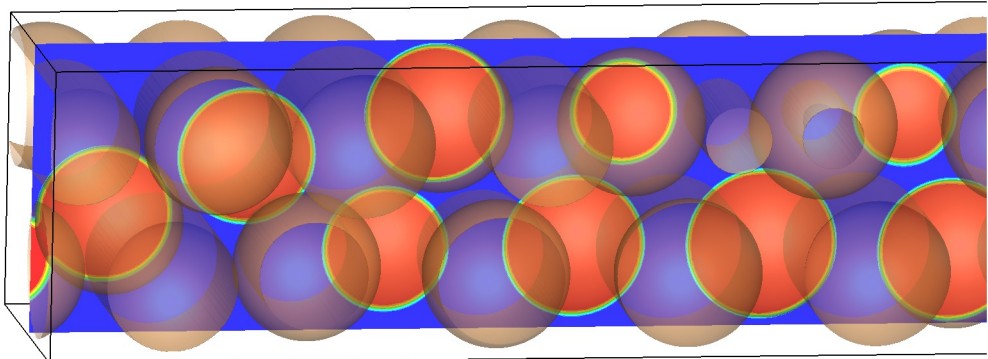

**Fig 1. Bubbles in a liquid illustrating multiphase flows in a periodic 3-D domain.** Only a small section of the domain is shown here.

**Table 1. Parameters relevant to the simulation setup we consider in this paper.**

| | |
|---|---|
| Domain | $8 \times 0.5 \times 0.5$ (see Fig 1) |
| Grid | $1600 \times 100 \times 100$ |
| Fluid densities | 1.0 (liquid) and 0.0001 (bubbles) |
| Boundary Condition | Periodic |
| Solver | Successive Over-Relaxation |
| Solver Tolerance | Relative Maximum Residual of $1e$-8 |
| Number of PEs | 200 |
| Domain Decomposition | 1-D along first dimension |

cluster uses Mellanox EDR interconnect. The MPI library chosen is Open MPI 4.0.1 compiled with gcc 8.3.0.

## 3 Baseline synchronous solver

Numerical iterative solvers of partial differential equations based on domain decomposition mostly involve two types of communication—(i) local communication of boundary values with neighboring PEs for computation of the boundary grid points (commonly known as halo exchange), and (ii) global communication of a convergence criterion among all the PEs for detection of the condition for termination. The traditional parallel programming paradigm in most numerical solvers is the bulk synchronous parallel [27] where all the PEs execute iterations in synchrony. This means that if some PEs are slow in their execution, all the other PEs have to wait for them to complete before moving to the next iteration together. In these solvers, the local communication with the neighboring PEs is usually done using MPI point-to-point two-sided communication routines `MPI_Send/Recv` [28]. The sending PE packs the boundary values into a message and invokes `MPI_Send` operation while the receiving PE receives and unpacks the message using `MPI_Recv` and copies it to augmented buffer points around its domain, popularly called *ghost cells*. The convergence detection involves global communication that is done using a collective communication routine called `MPI_Allreduce`. While the Allreduce routine aggregates the local convergence criterion from all the PEs to calculate the global convergence criterion, it also introduces a synchronization point at the end of every iteration, meaning that all the PEs have to start the next iteration together. The pseudo code for the synchronous solver is shown in Algorithm A.

The global synchronization and the two-sided MPI local communication often impose significant communication overhead which can affect the time and energy performance of the solver. Consequently, many improvements over the baseline algorithm have been suggested. One popular way is to overlap the communication with computation by replacing the blocking versions of communication routines with non-blocking versions [29]. This can be done for both the local and global communication. The non-blocking versions differ from their blocking counterparts in that the communication routine works in the background without pausing the code execution. For the local communication, the blocking versions `MPI_Send/Recv` can be replaced with non-blocking versions such as `MPI_Isend/Irecv`. While these non-blocking versions can save on time, they still require `MPI_Wait` at the end of every iteration to ensure that all non-blocking operations have completed. The `MPI_Wait` operation is also critical to ensure that the buffer used by the non-blocking operations is freed, otherwise memory leakage will occur. Similarly, the `MPI_Allreduce` for global convergence detection can be replaced by its non-blocking equivalent `MPI_Iallreduce` but `MPI_Wait` is still required. The `MPI_Wait` makes the PEs wait for each other before starting the next iteration —thus the solver stays synchronous. In Section 4, we describe an asynchronous algorithm that departs fundamentally from this bulk synchronous parallel paradigm.

**Algorithm A**: Baseline Bulk Synchronous Parallel Solver

```
1: do
2:    Compute values
3:    Communicate boundary values to neighbors using MPI two-sided
4:    Calculate Local Residual
5:    Calculate Global Residual using MPI collectives
6:    if Global Residual < Tolerance then
7:       Global Convergence detected
8:    end if
9: while Global Convergence not detected
```

## 4 Proposed asynchronous solver

To make a solver truly asynchronous, we propose a paradigm of parallel programming where the PEs do not wait for each other but rather execute computations with whatever values were last received from the other PEs. In this paradigm, there are no "global" iterations—rather every PE executes its own "local" iterations at its own pace without any global synchronization. Henceforth we use the term iteration to refer to local iterations of a PE which may differently progress for different PEs. The traditional two-sided MPI communication is not suitable for this purpose. Rather one-sided communication or Remote Memory Access is used [30, 31]. In one-sided communication, the sending PE can directly write into the memory of the receiving PE without the involvement of the receiver, unlike two-sided communication. Since no acknowledgement of communication is required from the receiver, there is no synchronization involved and thus one-sided communication is faster than two-sided communication. We note that Nayak et al. [32] also developed asynchronous solvers with MPI one-sided communication for domain decomposition but in the context of restricted additive Schwarz solvers. The restricted additive Schwarz solvers are two-level domain decomposition solvers [7] which is different from the simpler one-level SOR solver we consider here. An illustration comparing synchronous and asynchronous solvers is provided in Fig 2.

In one-sided communication, typically every PE defines a region of memory called *window* which is public [30]. This means that the other PEs have permission to access this window without the involvement of the PE—hence one-sided communication is often alternately called Remote Memory Access. The sending PE can invoke `MPI_Put` to directly write into this window of the receiving PE without the receiving PE's involvement. However, one-sided

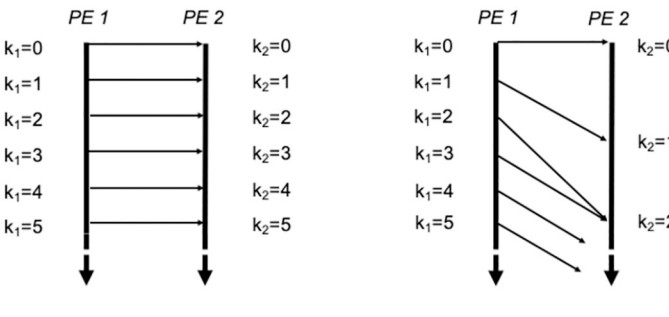

**Fig 2. Comparison between synchronous and asynchronous solvers between two PEs.** The vertical axis is wall clock time. The variable $k_i$ refers to the iteration count at the $i$-th PE. In the synchronous solver, every PE will execute the same iteration number at a certain point in time. In contrast, every PE in the asynchronous solver independently executes its iterations and may execute different iteration numbers at a certain point in time.

communication requires a mechanism to signal the beginning and end of an epoch of window access. These are of two kinds: (i) active, where the target is actively synchronized before its window can be accessed, and (ii) passive, where no active synchronization is required. We consider passive target synchronization using `MPI_Win_lock/unlock` to prevent active involvement of the receiver. In this scenario, the communication is considered to be complete once `MPI_Win_unlock` is called. If the receiver accesses data from the window before the `MPI_Win_unlock` is called, it might be inconsistent, meaning all the cells in the boundary might not correspond to the latest iteration at the sender. In order to prevent this, we copy the contents of the window to other arrays after `MPI_Win_unlock` and use these arrays for local computations at the receiver to maintain consistency.

There is no synchronization point in the asynchronous algorithm. Consequently there is no opportunity for collective communication with `MPI_AllReduce` for determining global convergence. Rather, the convergence detection needs to be performed in a distributed manner. Some approaches have been proposed for distributed convergence detection [33, 34]. However, they do not consider the situation in which a PE may have to restart iterations after temporary local convergence if there is a change in the values received from neighbors—this phenomenon is further explained in the next paragraph. We maintain the provision for restarting iterations in our asynchronous solver.

The algorithm for our asynchronous solver is specified in Algorithm B. The overarching difference from the synchronous Algorithm A is that there is no two-sided communication and collective communication. This implies that there is no need for synchronization among the PEs. The halo exchange with the neighbors is performed using `MPI_Put`. Because there is no `MPI_AllReduce` to aggregate the global convergence criterion to terminate iterations in all the PEs together, there has to be a different scheme to detect global convergence in an asynchronous manner. To do so, one PE is assigned as the Master to monitor global convergence. Each PE checks its local residual and compares it with the specified tolerance. If the local residual stays lesser than the tolerance for a certain number of iterations, the PE is considered to have locally converged and it sends that information to the Master. Checking the local convergence criterion for a range of multiple iterations instead of one iteration makes the algorithm robust to oscillations in the residual. It is important to note that even though a PE stops its iterations after local convergence, it should still keep on monitoring the values received from its neighbors. If there is a sudden change in the values received from neighbors, it means that values from a different source term has reached its domain. If that is the case, the PE is then

locally *unconverged* and made to restart iterations until it satisfies the local convergence criterion again. Finally, when the master detects that all the PEs (including the master itself) have locally converged, it recognizes that as global convergence and sends the global convergence flag to all the PEs so that they terminate the iterations. The solver is then considered to have converged. Note that different PEs will take different number of iterations in this asynchronous solver, unlike the synchronous one.

**Algorithm B**: Proposed Asynchronous Solver

```
1: do
2:   if Local Convergence not detected then
3:     Compute values
4:     Communicate boundary values to neighbors using MPI one-sided
5:     Calculate Local Residual
6:     if (Local Residual < Tolerance) for a range of iterations then
7:       Local Convergence detected
8:     end if
9:   else
10:     if New values from neighbors detected then
11:       Nullify Local Convergence
12:     end if
13:     if PE Not Master then
14:       Communicate Local Convergence information to Master
15:     else
16:       if Local Convergence of all PEs detected then
17:         Global Convergence detected—Communicate this to all PEs
18:       end if
19:     end if
20:   end if
21: while Global Convergence not detected
```

In order to demonstrate the performance gain due to the asynchronous solver over the synchronous solver, we consider the example described in Table 1. Table 2 shows a comparison between Algorithm A and Algorithm B in terms of the solution time and the global relative maximum residual. It is important to note that due to the absence of global communication (`MPI_AllReduce`), the global relative maximum residual is not calculated during every iteration of the asynchronous solver in Algorithm B. However, for a comparison with the synchronous solver in Algorithm A, we determine the global relative maximum residual for the asynchronous solver using just two `MPI_AllReduce` calls—one when the iterations start and the other after the iterations have stopped. From Table 2, we note that the global relative residual for both the solvers remain less than the tolerance (1e-8), indicating that the quality of solution is acceptable and similar with both the solvers. However, the asynchronous solver is about 6 times faster than the synchronous solver. Thus, the asynchronous solver has better performance. It is worth pointing out that this performance improvement is a result of removal of global communication as well as replacement of two-sided with one-sided communication.

**Table 2. Comparison of the performance of the synchronous solver (Algorithm A) and the asynchronous solver (Algorithm B) for the simulation setup specified in Table 1.** The asynchronous solver achieves the accuracy threshold in much lesser time.

| Solver Type | Time[s] | Global Relative Max Residual |
|---|---|---|
| Synchronous | 10346 | 9.12e-9 |
| Asynchronous | 1691 | 7.38e-9 |

## 5 Proposed event-triggered communication solver

In this section, we build on the asynchronous algorithm to present an *event-triggered* communication algorithm that significantly reduces the number of messages exchanged between the PEs. The asynchronous solver described in Algorithm B assumes that the communication of the boundary values with the neighbor PEs takes place at every iteration of that PE. The basic insight behind the event-triggered algorithm is that communication at every iteration may not be necessary. For instance, if the boundary values either do not change, or change in ways that are predictable without any further information from the sender, then the accuracy of the calculations at the intended receiver do not significantly degrade. In other words, a communication is only needed when triggered by an *event* at the sender (e.g., the boundary value changing from the previously communicated value by more than a threshold). A solver employing such *event-triggered* communication is schematically compared to the asynchronous solver in Fig 3. There is some flexibility in defining the events. For concreteness, in this paper, we design the events based on the norm of the boundary values at the sender PE. When this norm changes from the norm at previous communication by more than a specified threshold, the boundary values are communicated to the PE that is the intended receiver. The ghost cells at the receiver are updated only upon communication (possibly with a delay imposed by the MPI one-sided communication). If values are received in the ghost cells at the receiver due to an event of communication triggered at the sender, the receiver uses those values for its computation. Otherwise, if new values are not received at the ghost cells, the receiver uses values that are extrapolated from previously received values for its computation. Thus the event-triggered communication rule can be summarized as follows:

**At Sender**: Send boundary values if the condition

$$|\text{Current Norm} - \text{Last Communicated Norm}| \geq \text{Threshold}$$

holds; otherwise do not send.

**At Receiver**: Use ghost cell values if new values are received in ghost cells; otherwise use extrapolated values based on previous ghost cell values.

Specifically, to track the changes in the boundary values, we compute the L-1 norm of the boundary value vector (by summing the absolute values) and compare it with the L-1 norm when the sending PE last sent its boundary value vector. If the absolute difference between the

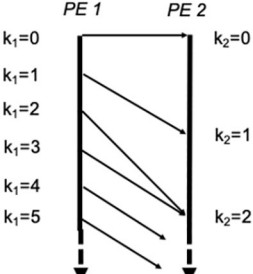 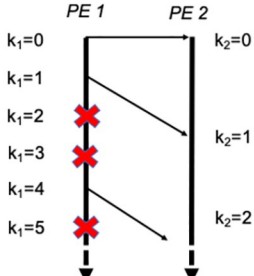

**Fig 3. Comparison between asynchronous solver and event-triggered solver as illustrated using two PEs.** The vertical axis is wall clock time. The asynchronous solver communicates at every iteration whereas the event-triggered solver communicates only when the event condition is satisfied. At other iterations, it avoids communication as shown by the red cross signs on the sender side.

two norms exceeds a certain threshold, an event of communication is *triggered*. Selecting the threshold is important for the overall efficiency and is a designer specified parameter. A low fixed value of the threshold will likely not result in much communication savings; however, it ensures that the solution with event-triggered communication closely tracks the solution with regular communication, especially if the boundary values are rapidly changing. On the other hand, a high fixed value of threshold will result in events being infrequently triggered, leading to communication savings but the solution with event-triggered communication may not track the one with regular communication, especially if the solution is changing slowly (e.g. close to the convergence). To obtain the best of both worlds, we propose an adaptive threshold policy that changes during the course of iterations of the solver. The solution is likely to rapidly change during early iterations in the solver, as high wavenumbers components of the error are eliminated. Therefore the boundary values also rapidly change, making a high value of threshold suitable in this region. However, during the later iterations when the solution starts approaching its final value, the boundary value slowly changes. In this situation, the threshold should be decreased to ensure that communication happens at least once in a while to reach the correct solution. To select the threshold based on how rapidly the solution is changing, at each PE, we compute the rate of change (or slope) of the norm of the vector of boundary values as the difference between the current norm and the norm at the last communication, divided by the number of iterations at the PE since then. The rate of change is then multiplied by a designer specified parameter $h$ called *horizon* to set the threshold $\tau^*$. Intuitively, the horizon $h$ signifies the number of iterations to look ahead while calculating the threshold. This is schematically shown in Fig 4(a) between two events $E1$ and $E2$. It is important to note that in this idea, the threshold would stay constant between two events. Due to this, there can be situations when the time between events can become excessively long. As an extreme case, we look at Fig 4(a) where the absolute difference of the norm of the boundary never crosses the threshold $\tau^*$ after event $E2$. This means that no further events of communication will be triggered which is detrimental to the convergence of the solver. To prevent this phenomenon, we modify the above threshold by adding a term for gradual decay. As shown in Fig 4(b), we define a *decay* parameter $d$, where $0 < d < 1$ is a user selected parameter. At every iteration, if no communication event happens, the threshold for the next iteration is decreased by multiplying the current threshold by $d$. Thus the threshold after $m$ iterations since the last transmission (which is

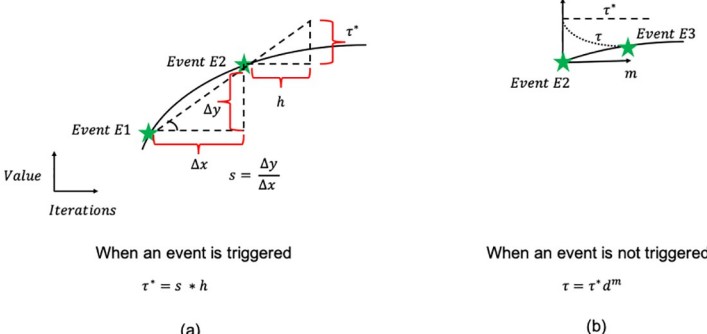

**Fig 4. Procedure for calculation of the threshold of event-triggered communication at the sending PE.** When the event E2 is triggered, a new threshold $\tau^*$ is calculated by multiplying the local slope $s$ between events E1 and E2 with the horizon $h$ as shown in the left subfigure (a). Then the previously calculated threshold $\tau^*$ is gradually decayed in the form $\tau = \tau^* d^m$ where $0 < d < 1$ is the decay rate and $m$ is the number of iterations since the event E2 when $\tau^*$ was calculated. This decay phenomenon, shown in subfigure (b), continues until the next event E3 triggers.

event $E2$ in Fig 4(b)) is given by $\tau = \tau^* d^m$. This decay continues until the next communication event $E3$ happens. During event $E3$, the threshold is again set to a new value of $\tau^*$, and then decayed similarly.

In our experiments, we observed that during the first few thousand iterations, the solution significantly oscillates. Since we want these oscillations to die out soon, we decided to communicate at every iteration for these first few thousand iterations, i.e., without invoking the event-triggered communication rule. However, this number is small compared to the total number of iterations required until convergence. For our experiments, we fix this initial number of iterations to be 2000.

As mentioned before, the receiver PE uses extrapolated values for its computation if new values are not received in the ghost cells. In order to perform the extrapolation, it stores a history of previously received values. The length of the history would obviously depend on the order of extrapolation. In this paper, we assume linear extrapolation although higher order extrapolation may be possible. The extrapolation is subject to certain considerations. In order to understand that, we look at the two scenarios when the receiver PE does not receive a message. First when the corresponding sender PE has not locally converged and does not send a message since the event criterion is not satisfied during that iteration and secondly when the corresponding sender PE has locally converged and hence stops sending messages. The extrapolation should be done for the former but not the latter. In other words, the extrapolation should only be done when the sending PE is still executing iterations and expects to send a message within the next few iterations. To distinguish between the two scenarios, it is important for the sending PE to send the local convergence flag to its neighbors in addition to the master. The pseudo code for the event-triggered communication algorithm is provided in Algorithm C. The major change from Algorithm B is the event-triggered halo exchange section specified in lines 4–13 and the communication of local convergence information to neighbors in Line 22.

**Algorithm C**: Proposed Event-Triggered Communication Solver

```
1: do
2:   if Local Convergence not detected then
3:     Compute values
4:     if Change in Boundary Values > Threshold then
5:       Send boundary values to neighbors using MPI one-sided
6:     end if
7:     if New values from neighbor received then
8:       Copy new values to ghost cells
9:     else
10:        if Neighbor not locally converged then
11:          Extrapolate ghost cell values based on history
12:        end if
13:     end if
14:     Calculate Local residual
15:     if (Local residual < Tolerance) for a range of iterations then
16:       Local Convergence detected
17:     end if
18:   else
19:     if New values from neighbors detected then
20:       Nullify Local Convergence
21:     end if
22:     Communicate Local Convergence information to neighbors
23:     if PE Not Master then
24:        Communicate Local Convergence information to Master
```

```
25:     else
26:       if Local Convergence of all PEs detected then
27:        Global Convergence detected—Communicate this to all PEs
28:       end if
29:     end if
30:   end if
31: while Global Convergence not detected
```

## 6 Simulation results

We now present our experimental results with the event-triggered communication algorithm. Fig 5 shows the L-1 norm of the left boundary values for some PEs, for the case specified in Table 1, using a horizon $h = 200$ and decay $d = 0.8$. Note that the x-axis of the plot in Fig 5 starts from 2000 since the event-triggered communication starts after the first 2000 iterations to wait for large scale oscillations to die out as mentioned before in Section 5. Now according to the algorithm, the boundary values are sent to the corresponding receiver only when the

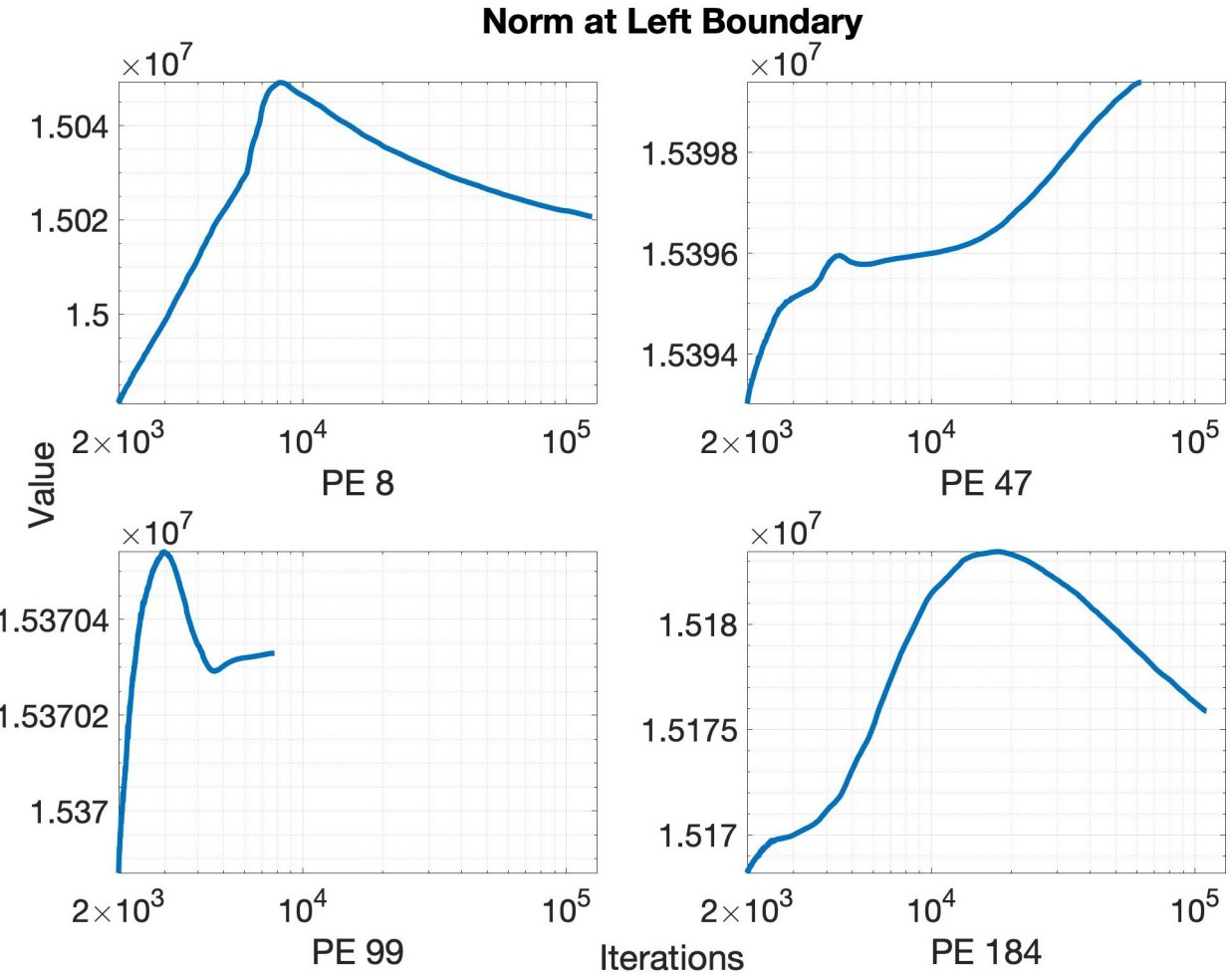

**Fig 5. Evolution of the Manhattan or L-1 norm of the top boundary of 4 randomly chosen PEs.** Note that the x axis starts from 2000 to wait for the large scale oscillations to die out.

## Threshold at Left Boundary

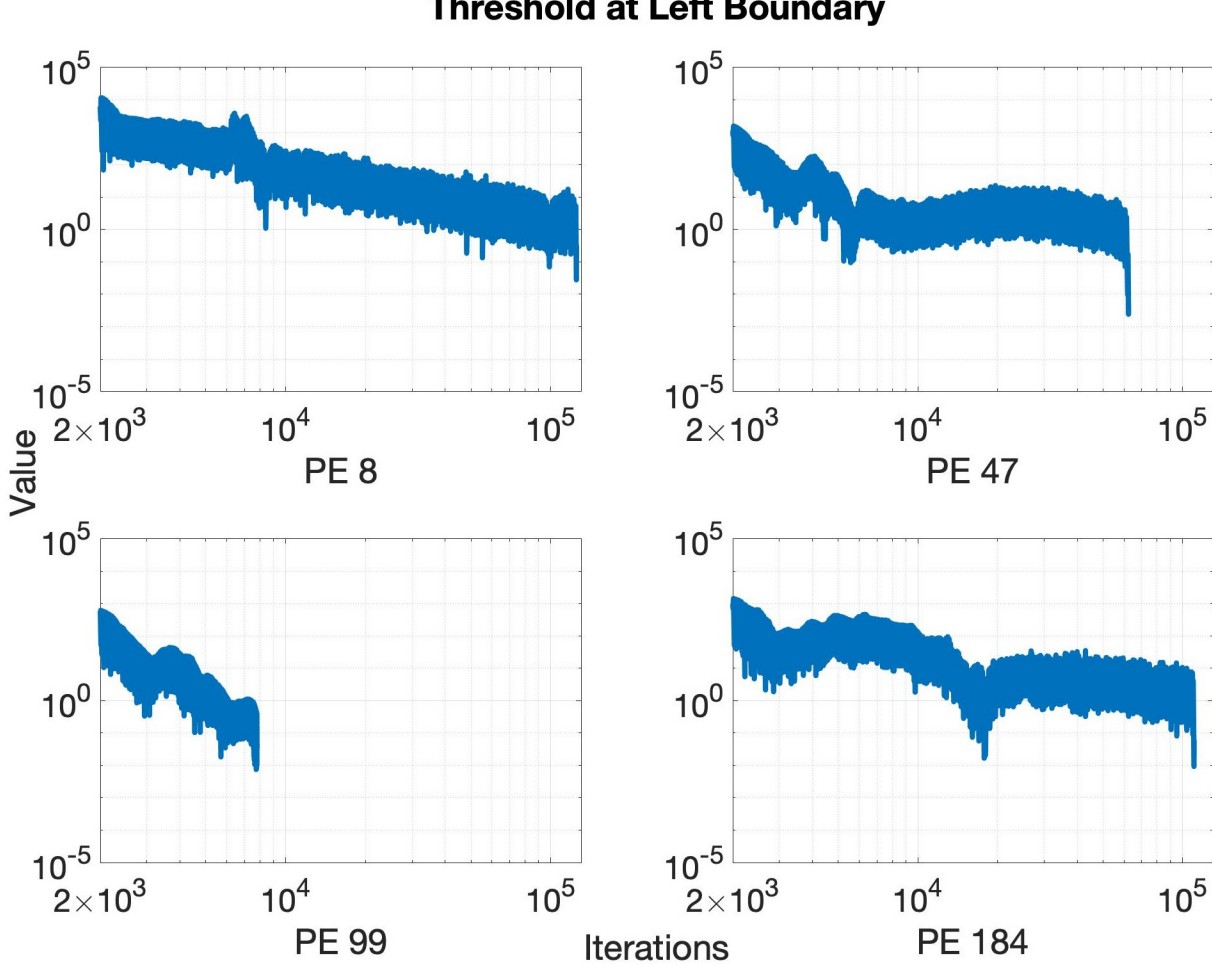

**Fig 6. Corresponding thresholds in a semi-log plot for the boundaries shown in Fig 5.** It is seen that the thresholds overall decrease with iterations to reflect the decrease in slope of the norm of the boundaries in Fig 5.

change in the norm exceeds the threshold. The corresponding thresholds for the boundaries in Fig 5 are shown in Fig 6. Since the slope of the boundary values decreases with time, the threshold also decreases to follow the trend of evolution of the boundary. The oscillations in the threshold seen in Fig 6 originate and are amplified by the local minor oscillations in Fig 5 that arise from the stochastic implementation delays of MPI one-sided communication. To reduce the effect of those oscillations on the threshold the sender PE keeps a history of multiple previously communicated events (instead of just one event as shown in Fig 4) and calculates the average slope. This average slope is then multiplied by the horizon to obtain the threshold. The length of the history is another user-controlled parameter, similar to the length of a moving average filter. Longer history results in a smoother slope, but at the cost of increased computational complexity. In this paper, we consider the length of this history to be 20 for our simulations.

Various performance metrics for the event-triggered communication algorithm are shown in Figs 7 to 9. Each simulation corresponding to a certain data point in these plots is run for 3 times and then the mean of them is plotted to account for the stochasticity of MPI one-sided

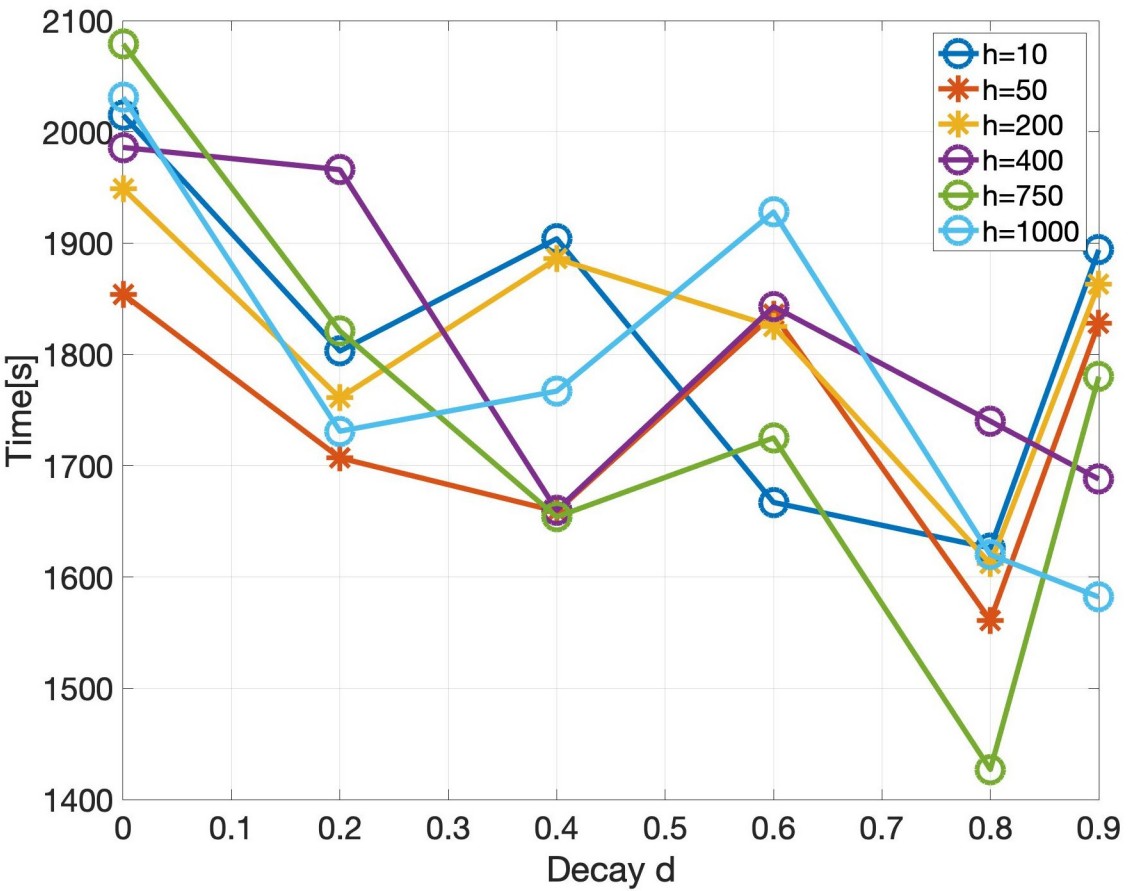

**Fig 7. Plot of time for simulation vs the decay *d* for various values of horizon *h*.** The decay and horizon are parameters that determine the event-triggered communication threshold.

communication. As a reminder, the parameters used in the experiments are specified in Table 1. The effect of the decay *d* and horizon *h* on the total simulation time is shown in Fig 7. Note that the values of time fluctuate a lot over different decay and horizon parameters due to the stochastic effects of MPI one-sided communication. However, we see an overall trend that as the decay and horizon is increased, the total time is reduced. In addition to reducing the total time, the event-triggered communication algorithm reduces the number of messages passed between the PEs. Fig 8 shows the reduction in the number of messages for various decay and horizon parameters, where the reduction is expressed as a percentage of the number of messages sent without event-triggered communication. We see that the percentage of messages drastically decreases with increasing decay *d* as well as increasing horizon *h*. This reduction in messages can not only lead to a decrease in simulation time as seen before, but also a decrease in energy consumption and congestion in interconnects. Note that it is difficult to measure the reduction in energy consumption directly but we point to a metric in [10] which states that around 1-3 pJ is spent in moving 1 bit of data between processors connected in a network. We also refer to [35] which highlights the aspect of decrease in congestion with reduction in messages. As a reminder, the quality of the solution with each of these event-triggered communication demonstrations is similar to that of the baseline synchronous solver since the global relative maximum residual (introduced in Table 2) is lesser than the specified

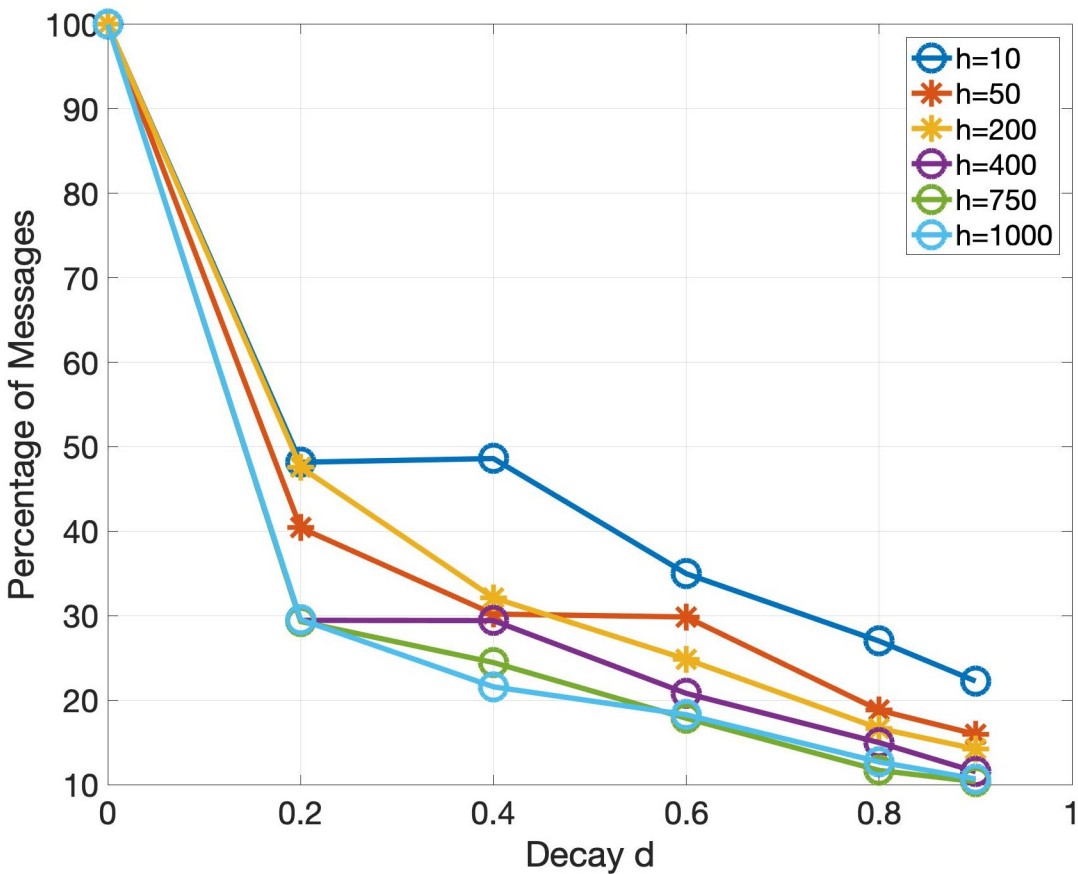

**Fig 8. Plot of the percentage of messages in the event-triggered communication solver for various values of horizon *h* and decay *d*.** Note that a decay of 0 is used to represent 100% of the messages since an event of communication is triggered for this case at every iteration. We see that as the decay and horizon increases, the percentage of messages starts to decrease.

tolerance of 1e-8. In Fig 9, we take a closer look at the reduction in number of messages for one particular simulation by plotting the number of messages triggered in each of the 200 PEs. The number of iterations that every PE takes to convergence with event-triggered communication is different as expected and depends upon the characteristics of the sub-domain assigned to that particular PE. However, the number of iterations taken in any PE is much lesser than that with the baseline synchronous solver. Further, the number of messages exchanged with event-triggered communication is even lower, highlighting the benefits of our algorithm.

## 7 Discussion

Communications between processing elements have always been a major concern with parallel computing. Thus algorithms that reduce the need for communications are likely to be needed in scientific and engineering simulations. Here, we first show that an asynchronous algorithm to solve the pressure Poisson equation encountered in numerical simulations of incompressible multiphase flows can significantly decrease the time to solution while maintaining similar accuracy. Then we develop another algorithm based on event-triggered communications that can further reduce the number of messages exchanged to solve that equation. This can reduce the overhead associated with communication, while maintaining the quality of solution.

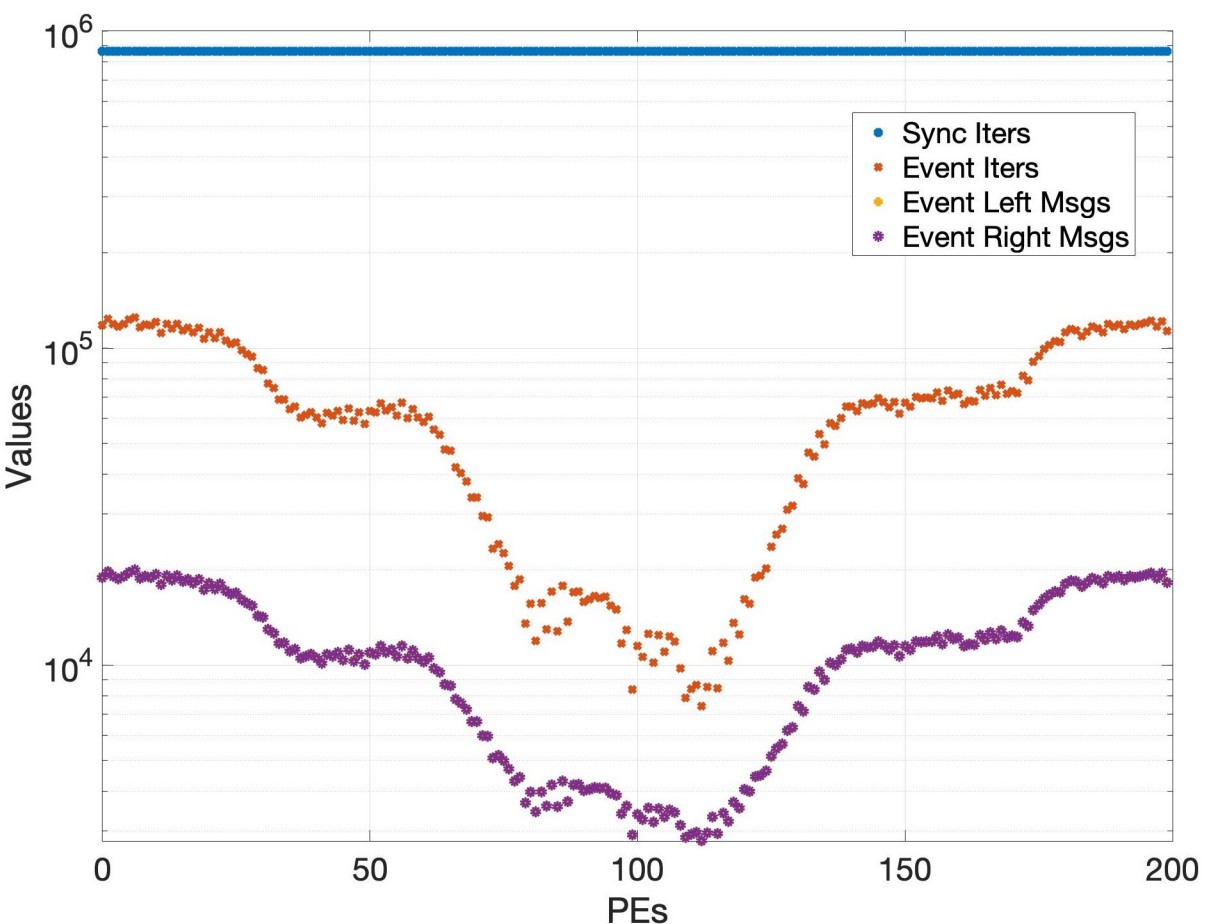

**Fig 9. Plot highlighting the number of messages sent by each of the 200 PEs to the left and right neighbors with the event-triggered communication algorithm considering horizon _h_ = 750 and decay _d_ = 0.8.** The number of iterations with the synchronous solver (Sync Iters in the plot legend) which is the same for all PEs is shown by the blue line for reference. In contrast, the number of iterations taken by each of the PEs in the event-triggered solver (Event Iters in the plot legend) is shown by the red asterisks. Further, the number of messages sent to the left neighbor and right neighbor (shown as Event Left Msgs and Event Right Msgs in the plot legend) by each of the PEs is shown respectively by the yellow star sign and the purple round sign. The number of messages for both the left and right neighbors for every PE are quite close. Hence the purple round signs overlap the corresponding yellow star signs for all the PEs. It is seen that the number of messages is considerably lesser than the number of iterations for any PE, thus illustrating the benefit of reduced messages in event-triggered communication.

Although the algorithms introduced here have been implemented using a very simple SOR solver, we believe that the strategy carries over to more sophisticated solvers, although the exact savings will, of course, be different.

## Author Contributions

**Conceptualization:** Vijay Gupta, Gretar Tryggvason.

**Formal analysis:** Vijay Gupta.

**Funding acquisition:** Vijay Gupta, Gretar Tryggvason.

**Methodology:** Soumyadip Ghosh, Jiacai Lu, Gretar Tryggvason.

**Project administration:** Vijay Gupta.

**Software:** Soumyadip Ghosh.

**Supervision:** Jiacai Lu, Vijay Gupta.

**Writing – original draft:** Soumyadip Ghosh.

**Writing – review & editing:** Vijay Gupta, Gretar Tryggvason.

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
