## [Decision Letter · Decision Letter 0]

28 Aug 2022

PONE-D-22-19810Communication-Efficient Algorithms for Solving Pressure Poisson Equation for Multiphase Flows using Parallel ComputersPLOS ONE

Dear Dr. Gupta,

Thank you for submitting your manuscript to PLOS ONE. After careful consideration, we feel that it has merit but does not fully meet PLOS ONE’s publication criteria as it currently stands. Therefore, we invite you to submit a revised version of the manuscript that addresses the points raised during the review process. You are invited to subtentially revise this work based on the reports (see below), and respond to all concerns in a clear manner. 

We look forward to receiving your revised manuscript.

Kind regards,

Mohamed Kamel Riahi

Academic Editor

PLOS ONE

Journal Requirements:

“This research was supported in part by the University of Notre Dame Center for Research Computing through its computing resources. The work of the authors was supported in part by NSF CBET-1953090 and CBET-1953082.”

“This research was supported in part by the University of Notre Dame Center for Research Computing through its computing resources. The work of the authors was supported in part by NSF CBET-1953090 and CBET-1953082.”

3. Please expand the acronym “NSF” (as indicated in your financial disclosure) so that it states the name of your funders in full.

“This research was supported in part by the University of Notre Dame Center for Research Computing through its computing resources. The work of the authors was supported in part by NSF CBET-1953090 and CBET-1953082.”

Additional Editor Comments:

Reviews have been received for the manuscript entitled "Communication-Efficient Algorithms for Solving Pressure Poisson Equation for Multiphase Flows using Parallel Computers".

The authors are invited to subtentially revise their work based on the reports, and respond to all concerns in a clear manner.

Reviewers' comments:

Reviewer's Responses to Questions

**Comments to the Author**

1. Is the manuscript technically sound, and do the data support the conclusions?

Reviewer #1: Partly

Reviewer #2: Partly

2. Has the statistical analysis been performed appropriately and rigorously? 

Reviewer #1: Yes

Reviewer #2: No

3. Have the authors made all data underlying the findings in their manuscript fully available?

Reviewer #1: Yes

Reviewer #2: No

4. Is the manuscript presented in an intelligible fashion and written in standard English?

Reviewer #1: Yes

Reviewer #2: Yes

5. Review Comments to the Author

Reviewer #1: Authors proposed two parallel algorithms to solve the pressure Poisson equation encountered in numerical simulations of incompressible multiphase flows. The first one is an asynchronous algorithm that removes the requirement of synchronization. The second is a event-triggered communication algorithm that reduces the number of messages while maintaining similar accuracy of solution. The paper is easy to read and is well structured.

However,

1. Reading the authors' previous work, I feel that this work lacks novelty. So the first question I have for them is this: what is the difference between this work and your previous work conceptually?

2. In the experimental results, authors said :

"This reduction in messages can not only lead to a decrease in simulation time as seen before, but also a decrease in energy consumption and congestion in interconnects."

How did you measure energy consumption and congestion? What metrics did you use? Where are the curves that actually show that energy has been reduced ?

3. In the experimental results, authors said :

"The effect of the decay d and horizon h on the total simulation time is shown in Fig 7."

How many processing elements did you use to get these results? This information was not specified.

4. What are the complexity your algorithms with regard to total execution time and number of communication rounds ? What is the different between the complexity of this work and your previous works ?

5. In section "Proposed Event-Triggered Communication Solver", authors said :

"When this norm changes from the norm at previous communication by more than a specified threshold, the boundary values are communicated to the PE that is

the intended receiver."

Can there be interlocking situations? And what happens during a collective communication?

6. In section "Proposed Asynchronous Solver", authors said :

"We note that other researchers [30] also developed asynchronous solvers with MPI one-sided communication for domain decomposition but in the context of restricted additive Schwarz solvers which is different from the SOR solver we consider"

Can you clearly show us the difference ?

7. In the introduction, authors said :

"However, these s-step methods considered parallelization using operator decomposition which is different from the parallelization using domain decomposition considered in this paper."

Can you clearly show us the difference ?

I have some suggestions :

1. In the whole paper, replace "Pressure Poisson" by "pressure Poisson".

2. Rewrite the last paragraph of the introduction. Authors said :

"In section 5, we extend the solver by adding event-triggered communication. In both sections 4 and 5, we present results for the respective algorithms."

It is clear that there is a problem.

3. Rename the section "Experimental Results" to "Simulation Results"

4. Perform other simulations by varying the number of processing elements and present the performance.

5. Discuss the results archieved throughout the paper in the conclusion and give perspectives of this work.

6. Add the URLs or DOIs of references. Also, authors must review the reference [11] and [21] (they cannot put the preprint a paper that have been published).

7. I have noticed that authors have some problems with the natural position of adverbs. There are also some little mistakes :

page 3:

and show that it reduces the computation time significantly. => and show that it significantly reduces the computation time.

Message Passing Interface (MPI). => message passing interface (MPI).

a simple in-house successive over-relaxation (SOR) solver => a simple in-house SOR solver

page 4:

SOR solver using domain decomposition domain decomposition => SOR solver using domain decomposition

page 6:

In the following section, we describe an asynchronous algorithm

that departs fundamentally from this bulk synchronous parallel paradigm. => In Section 4, we describe an asynchronous algorithm that fundamentally departs from this bulk synchronous parallel paradigm.

which may progress differently for different PEs. => which may differently progress for different PEs.

the sending PE can write directly into => the sending PE can directly write into

We note that other researchers [30] => We note that Pratik et al. [30]

MPI_Put to write directly into => MPI_Put to directly write into

page 7:

solver executes its iterations independently => solver independently executes its iterations

page 9:

Comparison of the performance of the Synchronous Solver (Algorithm A) and Asynchronous Solver => Comparison of the performance of the synchronous solver (Algorithm A) and asynchronous solver

that reduces the number of messages exchanged between the PEs significantly. => that significantly reduces the number of messages exchanged between the PEs.

then the accuracy of the calculations at the intended receiver do not degrade significantly. => then the accuracy of the calculations at the intended receiver do not significantly degrade.

A solver employing such event-triggered communication is compared schematically to => A solver employing such event-triggered communication is schematically compared to

page 10:

if the boundary values are changing rapidly. => if the boundary values are rapidly changing.

in events being triggered infrequently => in events being infrequently triggered

to change rapidly => to rapidly change

also change rapidly => also rapidly change

value changes slowly. => value slowly changes.

page 11:

This is shown schematically => This is schematically shown

page 12:

oscillates significantly => significantly oscillates

the norm increases gradually => the norm gradually increases

page 14:

Figure showing the evolution of the Manhattan => Evolution of the Manhattan

Figure showing the corresponding thresholds in a semi-log plot => Corresponding thresholds in a semi-log plot

page 16:

messages decreases drastically with => messages drastically decreases with

page 17:

shown by the yellow star sign and the purple round sign respectively. => shown respectively by the yellow star sign and the purple round sign.

Reviewer #2: The manuscript presents a solver for the Poisson problem, which is an important step in the solution of many approximations to partial differential equations. In the present case, the simulation background is multiphase flows, where a Poisson equation with variable coefficients needs to be solved potentially many times. The main idea of the manuscript is to replace the classical two-sided MPI communication (MPI_Send/MPI_Recv) in PDE solvers by one-sided communication (MPI_Put), and make the local computations work on whatever information happens to be present from the neighbors at a specific instant in time. This approach is intimately linked to the solver type, which is a successive overrelaxation (SOR) method. This works because the information from neighboring sites will change slowly, and any inconsistency a few iterations back or forth will merely change the convergence path, but not the convergence overall. As a further improvement of the algorithm, the authors propose to make the data exchange conditional, based on the rate of change of the data, combined with some extrapolation.

This overall idea is interesting and deserves to be explored, which is why I would like to see this contribution to be eventually published. However, the current version of the manuscript is not in a state that should be published in a scientific journal, as it contains questionable results that cannot be reproduced (or understood) and because the reasoning is not at the level of the state-of-the-art. I therefore invite the authors to address the following major concerns:

- The problem to be analyzed is not sufficiently specified, making it impossible to understand what exactly gets done. More precisely, the authors need to state the exact version of the SOR algorithm, the form of the coefficients rho in the figure from Figure 1, the number of iterations, overlap/data dependency resolution in terms of the classical SOR algorithm in terms of the domain decomposition, and similar settings. Note that I tried to compile and run the code from the given repository, but the instructions are insufficient. I managed to figure out the command line arguments from the source code, but there is apparently need for some input arguments that are hardcoded to the author's computer, rather than available from the repository. The repository needs to contain a state the works on a third-party computer, at least for such a simple case with no NDA issues.

- Regarding the asynchronous algorithm of Section 4, I would like the authors to state how precisely the neighboring information is used for the local computations: In general, we need to assume that the neighbor might be in the process of writing to the place where the receiver is just reading from. Is there a way that invalid data could appear, and is this operation guaranteed to be using valid data? I believe yes, as the data will not get written in bit granularity, but cacheline granularity, and at the remaining gap cache coherency will ensure that anything that has arrived will be the long-term valid state.

- The overall motivation for choosing the simple algorithm SOR needs to be reframed. If I understand the results in Figures 5 and 9 correctly, the iteration count is between 100k and 1 million. For solving a Poisson problem, this is not competitive, and findings may be skewed due to this selection, see also below. While the authors mention this in the introduction (page 3, first full paragraph), the point is that even variable-coefficient Poisson problems can be solved at the equivalent of with 10-100 applications of the fine-level operator. This is achievable with multigrid methods, taking SOR smoothening and some 5-10 iterations. Similar costs can be obtained with BDDC or FETI-DP methods. To be specific, I believe this problem can be solved in around 1 second for 48 cores (1 node) of the given AMD architecture, which need to be compared to around 1700 seconds on 200 processing elements (cores?) according to Table 2. Obviously, 1 second versus 1700 seconds sets very different requirements on the communication timings, and this needs to be set in proper context. The authors mention hypre as a frequently used solver: I strongly suggest to add its timings; the objective will not be to beat it or even be close in timings (as hypre should be near optimal), it is needed to set a relevant anchor. Note that I do see value in keeping this simpler algorithm (as a first research step to gain insight), but the conclusions need nonetheless be anchored against the current state of the art.

- Regarding the previous comment, I definitely see value of the proposed ingredients for multigrid smoothers, so at least section 4 seems to be very relevant for a moderate number of sweeps, say 5-10, given the right background. However, the event-based solver is questionable at the very least: The authors argue that the changes of boundary data gets less as the SOR iteration is advanced, advocating some extrapolation at later stages. If I interpret the results correctly, this is because the SOR iteration smoothens the error, leaving only low-frequency parts of the error, where the nearest-neighbor exchange is now between very similar values. However, that would be the place where one really needs to go to coarser grids (or a BDDC coarse space), because the work spent on a fine grid is really in vein: A lot of work, but extremely slow convergence. Without proper background and additional motivation, I cannot see how this would be useful for a practically relevant PDE solver, apart from educational purposes. Again, I emphasize that I do not dispute the value of event-based steps in general, but here the events are applied to a property that a state-of-the-art algorithm will not expose, because it has employed much more efficient steps.

- Coming to the quality of the results, the first data point given in Table 2 suggests a run time of 10,346 seconds for the synchronous solver and 1,691 seconds for the proposed solver, which the authors credit to the one-sided MPI communication. In my experience, the difference is way too high, and I would rather expect the advantage to be in the range of 5-10% in the best case [*]. Since I am not able to run the authors' code, my impression is that something in the experiment is broken. To gain understanding, it would be important to list the number of iterations in the synchronous case and compare that number to some relevant statistical average of the number of local iterations across all 200 PEs in the asynchronous case. If these two numbers different substantially, the one-sided MPI communication is not the cause, and something similar could then also be emulated with MPI_Send/Recv. Or put differently, if the iteration count explains the difference, the finding is useless and linked to a bad algorithm choice. Also beyond the question of iteration counts, to make a valuable scientific contribution the authors would need to explain the origin of the difference in timings, by identifying a bottleneck in the synchronous algorithm (in the sense of timings of a single iteration, and comparing the achieved throughput to the hardware capabilities), and showing how the one-sided MPI algorithm relaxes on this bottleneck. The authors vaguely talk about the cost of exchanging messages, but I am not convinced. This can be seen from timings: If we assume the synchronous solver runs 10k seconds and takes 10^6 (one million) iterations, the time per iteration is 0.01 seconds. Regular MPI ghost exchange scales well all the way to 1e-4 seconds on architecture similar to the one used here [**], so there must be something else, e.g. improper load balancing or other aspects.

- The authors state that they use 200 processing elements (PEs). I assume this means 200 MPI ranks. As the machine they use has nodes with 48 cores each, this means that some nodes are not fully populated. The exact mapping of work to cores and nodes needs to be specified, also regarding the possible load balancing problem. Generally, I strongly recommend to use a number of subdomains that completely fill a node. Also, to give a proper analysis of the algorithms, experiments on different numbers of PEs (nodes) would be very helpful, in the context of a strong scaling experiment.

- The one-sided communication adds a statistical component to the algorithm and its performance. This would demand for a statistical analysis of the results: If I execute the algorithm 10 times, I would expect different results, both in terms of iterations to convergence (here, iteration needs to be an average on its own as different subdomain will produce different numbers) and timings. The result needs to be shown with its statistical distribution, error bars around some expectations, and a proper discussion needs to be added.

- The presentation quality is rather poor overall, e.g. Figure 5 should use logarithmic scaling of the x axis, maybe also Figure 6. This is one example, other figures should also be looked at to provide a good presentation.

- I suggest the authors to tone down or skip their high-level discussion about HPC, microprocessors or parallelism in the introduction and conclusions. I feel that the authors do not really know what they are talking about, highlighting some tangentially relevant metrics they found in some "keynote" papers, but completely ignoring more mundane and more relevant problems with their algorithms and presentation. Without a sense for the absolute timings and bottlenecks, the drawn conclusions might or might not be what is relevant for state-of-the-art solvers. Note that I do not dispute the study of simplified problems or non-optimal algorithms for the sake of exploring fundamentally new ingredients, but it needs to be done against a proper understanding of the state-of-the-art with relevant links.

Footnotes:

[*] I could not find good recent PDE references for this property, because most HPC architectures and contributions from US HPC grounds are GPUs nowadays, where tradeoffs are different. One thing I found is the work by Munch et al., ACM TOMS 47(4), 2021, https://doi.org/10.1145/3469720, where MPI-3 shared memory is explored to reduce communication (Table 8 on page 23). In 6D, those authors report up to 30% speedup, so in 3D I would not expect more than 10% in terms of run time reduction if the domain is well-partitioned between processes.

[**] Again, it is not easy to find good references. I recommend the authors to check Kolev et al., Int J High Perf Comput Appl 35(6), 2021, https://doi.org/10.1177/10943420211020803, Figures 12 (CPU) and Figures 13 (GPU). For a simple setup, the authors might also want to consult Figure 3 of Kronbichler and Wall, SISC 40(5), 2018, for a classical strong scaling curve with the current state of the art.

6. PLOS authors have the option to publish the peer review history of their article (what does this mean?). If published, this will include your full peer review and any attached files.

Reviewer #1: **Yes: **LACMOU ZEUTOUO Jerry

Reviewer #2: No

---

## [Author Response · Author response to Decision Letter 0]

12 Oct 2022

We have uploaded a file titled "Response to Reviewers" that considers each comment by the reviewers in detail.

---

## [Decision Letter · Decision Letter 1]

8 Nov 2022

Communication-Efficient Algorithms for Solving Pressure Poisson Equation for Multiphase Flows using Parallel Computers

PONE-D-22-19810R1

Dear Dr. Gupta,

We’re pleased to inform you that your manuscript has been judged scientifically suitable for publication and will be formally accepted for publication once it meets all outstanding technical requirements.

Kind regards,

Mohamed Kamel Riahi

Academic Editor

PLOS ONE

Additional Editor Comments (optional):

Reviewers' comments:

Reviewer's Responses to Questions

**Comments to the Author**

1. If the authors have adequately addressed your comments raised in a previous round of review and you feel that this manuscript is now acceptable for publication, you may indicate that here to bypass the “Comments to the Author” section, enter your conflict of interest statement in the “Confidential to Editor” section, and submit your "Accept" recommendation.

Reviewer #1: All comments have been addressed

2. Is the manuscript technically sound, and do the data support the conclusions?

Reviewer #1: Yes

3. Has the statistical analysis been performed appropriately and rigorously? 

Reviewer #1: Yes

4. Have the authors made all data underlying the findings in their manuscript fully available?

Reviewer #1: Yes

5. Is the manuscript presented in an intelligible fashion and written in standard English?

Reviewer #1: Yes

6. Review Comments to the Author

Reviewer #1: The authors have done an outstanding job of revising this article. The recommendations have been well applied. The paper is ready for publication !

7. PLOS authors have the option to publish the peer review history of their article (what does this mean?). If published, this will include your full peer review and any attached files.

Reviewer #1: No

---

## [Editor Report · Acceptance letter]

11 Nov 2022

PONE-D-22-19810R1 

Communication-Efficient Algorithms for Solving Pressure Poisson Equation for Multiphase Flows using Parallel Computers 

Dear Dr. Gupta:

I'm pleased to inform you that your manuscript has been deemed suitable for publication in PLOS ONE. Congratulations! Your manuscript is now with our production department. 

Kind regards, 

on behalf of

Dr. Mohamed Kamel Riahi 

Academic Editor

PLOS ONE